# Hepatocellular Carcinoma Growth Kinetics and Outcomes After Transarterial Embolization: A Single-Center Analysis

**DOI:** 10.3390/cancers17203346

**Published:** 2025-10-17

**Authors:** Ken Zhao, Harrison Blume, Elena N. Petre, Dimitrios Xenos, Zuzanna Kobus, Erica S. Alexander, Vlasios S. Sotirchos, Ruben Geevarghese, Anne Covey, Joseph P. Erinjeri, Etay Ziv, Constantinos T. Sofocleous, James J. Harding, Kevin Soares, Carlie Sigel, Efsevia Vakiani, Hooman Yarmohammadi

**Affiliations:** 1Department of Radiology, Memorial Sloan Kettering Cancer Center, New York, NY 10065, USA; blumeh@mskcc.org (H.B.); xenosd@mskcc.org (D.X.); alexane@mskcc.org (E.S.A.); sotirchv@mskcc.org (V.S.S.);; 2Department of Medicine, Memorial Sloan Kettering Cancer Center, New York, NY 10065, USA; 3Department of Surgery, Memorial Sloan Kettering Cancer Center, New York, NY 10065, USA; 4Department of Pathology, Memorial Sloan Kettering Cancer Center, New York, NY 10065, USA

**Keywords:** hepatocellular carcinoma, transarterial embolization, growth kinetics

## Abstract

Transarterial embolization is an established liver-directed therapy for the treatment of hepatocellular carcinoma. Tumor growth rate may be associated with outcomes after transarterial embolization. It is also unknown if there exists a genetic mutation associated with the growth rate of hepatocellular carcinoma. This study aimed to determine if there is an association between tumor growth rate, overall survival, and tumor progression after transarterial embolization, as well as if any common genetic mutations are associated with tumor growth rate, in patients with hepatocellular carcinoma. Identification of prognostic factors may improve the treatment selection and counseling for patients with hepatocellular carcinoma.

## 1. Introduction

Hepatocellular carcinoma (HCC) is the third-leading cause of cancer-related mortality worldwide [1]. The diagnostic and therapeutic workup of patients with HCC is multidisciplinary, and treatment decisions are individualized based on factors such as underlying liver function, stage of disease, and performance status [2,3,4]. Curative approaches for the treatment of HCC have primarily involved surgical resection, liver transplantation, and percutaneous ablation, though emerging data supports curative potential with transarterial radioembolization (TARE) for early-stage unresectable HCC when performed using the radiation segmentectomy technique [5,6].

Most patients with HCC cannot be offered curative-intent therapy and are treated with the intent of disease control. Transarterial embolization (TAE) is a locoregional therapy that may be a first-line option in patients with liver-limited HCC who cannot be offered curative therapy [2,3,4]. The efficacy of TAE is comparable to transarterial chemoembolization (TACE) [7,8,9].

Tumor growth rate may be a prognostic factor for HCC patients treated with TAE. Existing studies on HCC doubling time demonstrate wide variability in growth rates, and there is limited retrospective data suggesting that rapid growth is associated with worse prognosis after liver-directed therapy [10,11,12]. Evidence for the prognostic value of HCC’s growth rate is limited partly because patients typically receive treatment shortly after diagnosis. Intraprocedural cross-sectional imaging obtained during TAE, such as computed tomography (CT) performed using a combined CT/angiography suite or rotational cone-beam CT, provides an opportunity to perform follow-up measurements of a tumor at the time of treatment and assess the growth rate.

There have also been efforts to identify a prognostic gene signature for patients with HCC. However, the existing data is sparse, largely because HCC is often diagnosed without tissue sampling [13,14]. Limited evidence suggests that a TP53 mutation may be associated with worse prognosis and earlier local progression in HCC patients treated with TAE or TACE [15,16]. It remains unknown if there is a mutation associated with HCC growth rate.

The objectives of the present study are (1) to evaluate the associations between tumor volume doubling time (TVDT) and clinical outcomes after TAE for patients with HCC and (2) to evaluate the impact of tumor genotype on TVDT. TVDT may be a prognostic factor for patients with HCC treated with TAE.

## 2. Materials and Methods

### 2.1. Patient Selection

This is an institutional review board-approved, single-institution study. Adult patients were identified via a prospective biospecimen protocol or a retrospective database. All components of data collection and analysis were compliant with the Health Insurance Portability and Accountability Act.

Patients prospectively enrolled between January 2014 and June 2022 underwent concurrent liver biopsy during TAE. All provided informed consent for data collection, which, along with biospecimen use, was approved by the institutional review board. The objectives of this current study were not the focus of the prospective protocol.

The retrospectively identified patients were treated with TAE between January 2014 and June 2022 and had tumor tissue obtained, either prior to or after TAE, via a liver biopsy or surgical resection. A separate institutional review board protocol approved the retrospective study, and consent was waived due to its retrospective nature.

The cohort included patients with tissue-proven HCC who underwent baseline contrast-enhanced cross-sectional imaging (CT or MRI) and subsequent TAE. The index tumor was defined as the largest tumor on initial cross-sectional imaging that was treated with TAE. Patients were excluded if the greatest trans-axial diameter of the index tumor was less than 1 cm, the index tumor was poorly visualized on initial or follow-up cross-sectional imaging, or they had HCC treatment other than TAE prior to the first cross-sectional follow-up imaging after TAE.

If a patient had follow-up contrast-enhanced CT or MRI prior to any HCC treatment, and this imaging was performed at least 30 days after the initial baseline imaging, measurements to determine TVDT were performed based on diagnostic cross-sectional imaging alone. Otherwise, intraprocedural imaging obtained during TAE, either CT performed using a combined CT/angiography suite or rotational cone-beam CT, was used to visualize the index tumor and obtain follow-up measurements to determine TVDT. Patients were excluded if the index tumor visualization on intraprocedural imaging was insufficient for accurate measurements or if TAE was performed less than 30 days after initial baseline imaging.

### 2.2. Tumor Volume Doubling Time Calculation

The initial volume of the index tumor on baseline imaging was determined based on trans-axial measurements of diameter on multiphasic contrast-enhanced cross-sectional imaging. Follow-up trans-axial measurements of the index tumor were obtained from either intraprocedural cross-sectional imaging acquired during locoregional therapy or subsequent diagnostic cross-sectional imaging acquired prior to locoregional therapy. These measurements were made by a 4th-year medical student (H.B.) and an attending board-certified interventional radiologist (K.Z.). All trans-axial tumor measurements were reviewed by K.Z. and adjusted if necessary.

Intraprocedural imaging used for follow-up tumoral measurements consisted of contrast-enhanced or non-contrast intraprocedural cross-sectional imaging, either CT as part of a combined angiography/CT suite or cone-beam CT with axial reconstructions. Intraprocedural contrast-enhanced cross-sectional imaging was performed prior to embolization to delineate vascular anatomy and tumoral perfusion. Intraprocedural non-contrast cross-sectional imaging acquired after embolization was performed to assess completeness of the embolization procedure via tumoral staining (Figure 1).

Tumor volume was calculated using the largest perpendicular trans-axial diameters with the assumption that the tumors were an oblate ellipsoid. The tumor volume doubling time was calculated using the Schwartz equation, with T representing the time point and V representing the volume [10]:
$$TVDT\;=\frac{\left(T-T_{0}\right)ln2}{\ln\left(\frac{V}{V_{0}}\right)}$$

### 2.3. Genetic Analysis

Tumor tissue specimens were obtained from percutaneous core needle biopsies or surgical resections. Specimens were fixed with formalin and embedded in paraffin following collection and were then examined by pathologists at the study institution. Genetic mutation analysis was completed using IMPACT, a hybridization-capture-based, targeted next-generation sequencing array [17]. Individual genetic mutations were included in subsequent analyses if present in at least 10% of the patient cohort.

### 2.4. Transarterial Embolization

TAE procedures were performed as previously described [7]. Following catheter selection into the hepatic arterial vasculature, all identified branches supplying the tumor were treated with microparticles (Embosphere^®^ Microsphere; Merit Medical, South Jordan, UT, USA). Initially, smaller beads (40–120 µm or 100–300 µm in diameter) were used for distal occlusion before larger beads were used for more proximal vessel occlusion. The particle size gradually increased until stasis was achieved, defined by contrast injection filling the targeted vessel and persisting without washout for five cardiac beats. For tumors large in size (>10 cm in diameter), tumors with extensive vasculature, or tumors found to have an arteriovenous shunt, larger particles (i.e., 300–500 µm) were initially used.

### 2.5. Data Collection

Baseline clinical characteristics were collected through a retrospective review of electronic medical records or from existing, maintained databases. Variables included demographic information (age, sex, ethnicity), etiology of liver disease, initial tumor characteristics (histopathologic diagnosis, tumoral genetic alterations), history of previous HCC treatment, including surgery, locoregional therapies, and systemic treatment, Eastern Cooperative Oncology Group (ECOG) performance score [18], follow-up (date of death or last known follow-up), and laboratory values, including alpha fetoprotein (AFP).

Baseline cross-sectional imaging was evaluated for the number of HCC tumors present, the distribution of tumors (uni- or bilobar), the presence of macrovascular invasion (invasion of the portal vein, hepatic veins, or inferior vena cava), the presence of extrahepatic disease, and the trans-axial dimensions of the index tumor. Each patient’s Barcelona Clinic Liver Cancer (BCLC) stage was assessed from imaging and clinical characteristics [2].

### 2.6. Outcome Assessment

The primary outcome was overall survival (OS) with relation to TVDT and was calculated from the date of the first TAE procedure to the date of death or last-known contact with the patient, whichever came first. The secondary outcome was local tumor progression-free survival (LTPFS) with relation to TVDT.

After TAE, the initial cross-sectional imaging was contrast-enhanced CT or MRI 1 month following the procedure. Subsequently, imaging was obtained typically at an interval of 3 months. Two board-certified attending interventional radiologists (K.Z. and H.Y., with 4 and 12 years of experience, respectively) reviewed the imaging for local tumor progression, which was defined as recurrence or an increase in size of the enhancing tumor at a site that was previously treated with TAE. Discordances were resolved by consensus decision.

### 2.7. Statistical Analysis

Median follow-up was calculated using the reverse Kaplan–Meier method. Survival analysis was performed using the Kaplan–Meier method. To determine if TVDT was associated with OS or LTPFS, multiple threshold TVDTs were used to subgroup the patients for analysis, starting with the median TVDT and decreasing in half-month intervals. If multiple TVDT thresholds were found to be statistically significant, the optimal threshold was determined using the minimum *p*-value approach [19]. At each candidate cutoff, the data was split into two groups, and the log-rank test was performed to compare the outcomes between the groups. The threshold corresponding to the lowest *p*-value—the most statistically significant difference between the groups—was chosen as the optimal threshold.

Univariate analysis of prognostic factors associated with survival was performed using Cox regression. Factors that were statistically significant on univariate analysis were then subsequently utilized for multivariate Cox regression analysis.

The time to local tumor progression, with death or the last patient contact without progression as competing risks, was used to obtain a cumulative incidence function. The cumulative incidence of local tumor progression and sub-distribution hazard ratios were estimated using Fine and Gray competing risks regression.

After subgrouping the patients based on a threshold TVDT, differences in baseline characteristics, including the presence of a genetic mutation, were assessed by Fisher’s exact test for categorical variables. For continuous variables, normality was first assessed using the Shapiro–Wilk test. The student’s t-test was used if normality was confirmed. Otherwise, the Mann–Whitney–Wilcoxon test was used. A *p*-value lower than 0.05 was considered statistically significant. Competing risk analysis was performed using RStudio Build 735 (Boston, MA, USA) [20]. All other statistical analyses were performed using STATA version 17 (College Station, TX, USA).

## 3. Results

### 3.1. Patient Characteristics

A total of 505 patients were screened for potential inclusion in this study. A study flowchart with reasons for patient exclusion is provided in Figure 2. The final cohort included 54 patients (47 male, 7 female) with a median age of 69 (IQR 59–73) years. The median follow-up time was 74.3 months (95% CI: 53.1–88.9). The baseline patient and lesion characteristics are summarized in Table 1.

The imaging modality used for follow-up tumor measurements and the corresponding number of patients were as follows: intraprocedural non-contrast CT using angio/CT (35), diagnostic contrast-enhanced CT (9), intraprocedural non-contrast cone-beam CT (5), intraprocedural contrast-enhanced CT using angio/CT (2), diagnostic contrast-enhanced MRI (2), and intraprocedural contrast-enhanced cone-beam CT (1). The median time between initial and follow-up imaging for tumor measurements was 63.5 (IQR 31–81.75) days.

Based on the OS results described subsequently, the patients were sub-grouped based on a TVDT threshold of 2.5 months. The patients (n = 21/54, 38.9%) with HCC that exhibited a TVDT of ≤2.5 months were considered to have rapidly growing tumors, and the patients (n = 33/54, 61.1%) with HCC that exhibited a TVDT of >2.5 months were considered to have slowly growing tumors. There was no statistically significant difference between the patient subgroups with regard to baseline demographics and lesion characteristics.

### 3.2. Overall Survival Analysis

The overall cohort exhibited a median TVDT of 4.08 months (95% CI, 0.74–65.62) and had a median OS of 35.1 months (95% CI: 21.5–63.5) (Figure 3A). Analysis of OS after stratifying the patients per various TVDT thresholds demonstrated that the patients with a shorter TVDT, corresponding to more rapid tumor growth, had a significantly worse OS starting at a TVDT threshold of 3.5 months (*p* = 0.036) (Table 2, Figure 3B–F). In particular, a TVDT threshold of 2.5 months was associated with the lowest *p*-value of 0.011; the patients with HCC exhibiting a TVDT of ≤2.5 months had a median overall survival of 30.6 months (95% CI: 8.8–39.8) after TAE, whereas the patients with HCC exhibiting a TVDT of >2.5 months had a median overall survival of 77 months (95% CI: 21.0–90.1). This TVDT threshold of 2.5 months was utilized for subsequent analyses because it demonstrated the strongest statistical significance.

Univariate Cox regression analysis of the overall cohort showed that the ECOG performance score, the presence of a TP53 or TSC2 mutation, an AFP level > 200 ng/mL, and a TVDT ≤ 2.5 months were factors significantly associated with worse OS (Table 3). In subsequent multivariate analysis, a TVDT ≤ 2.5 months, an ECOG score of 1 or 2, and an AFP > 200 ng/mL remained statistically significant as predictors of worse OS (Table 4).

### 3.3. Local Tumor Progression Analysis

The overall cohort had a median LTPFS of 7.6 months (95% CI: 4.5–11.0) (Figure 4A). Univariate analysis of factors affecting LTPFS showed that an ECOG score of 1 or 2, the presence of a TP53 mutation, and the presence of >5 tumors on initial imaging were statistically significant predictors of worse LTPFS (Table 5). In subsequent multivariate analysis, only the presence of >5 hepatic tumors on initial imaging remained a significant predictor of worse LTPFS (Table 6). Multiple TVDT thresholds were analyzed, ranging from 1.5 to 4.0 months, and none were found to be a statistically significant predictor of LTPFS (Table 7, Figure 4B).

Competing-risk analysis also showed that TVDT was not significantly associated with local tumor progression. For instance, cumulative incidences of local tumor progression were 33.3% (95%CI: 12.4–54.3%) at 6 months and 57.1% (95% CI, 34.9–79.4%) at 12 months for tumors with a TVDT >2.5 months versus 43.7% (95%CI: 26.2–61.2%) at 6 months and 68.7% (95% CI, 52.1–85.4%) at 12 months for tumors with a TVDT ≤ 2.5 months (*p* = 0.719) (Figure 5).

### 3.4. Genetic Mutations and TVDT

A histogram of the genetic mutations present in the cohort is presented in Figure 6. The most common mutations were TERT (39 patients, 74%), CTNNB1 (20 patients, 37%), TP53 (19 patients, 35.3%), ARID1A (12 patients, 22.2%), AXIN1 (8 patients, 14.8%), TSC2 (7 patients, 13%), and CDKN2A (6 patients, 11.1%). There was no statistically significant difference in the incidence of any common genetic mutation between the rapidly growing and slowly growing tumor subgroups (Table 1). No statistically significant association between the etiology of liver disease and the tumor subgroups was found either.

## 4. Discussion

Patients with rapidly growing tumors, defined as a TVDT less than or equal to 2.5 months, had worse OS after TAE. A recent retrospective study by Kim et al. also reported that HCC patients with rapidly growing tumors, defined as a TVDT less than 2 months, exhibited significantly worse OS compared to patients with slower-growing tumors [10]. Their study consisted of an East Asian cohort and included HCC patients who received varying initial treatment modalities, including resection, liver-directed therapies, and conservative treatment. Of their cohort, 29.7% of the patients were treated with TACE, and none were treated with TAE. Similarly, a study by Chen et al. that included 97 East Asian patients with huge HCCs (diameter > 10 cm) found that patients with a tumor growth rate of >8.6%/month had worse OS after TACE [12]. In contrast, the current study had a North American cohort, including only one patient with HCC diameter >10 cm, and all patients were treated with TAE. Regardless, the prior studies and this current study support an association between rapid tumor growth and worse patient survival after transarterial locoregional therapy for treatment of HCC.

Patients in the present study had TVDTs consistent with the prior literature. The growth rate for HCC is known to be variable, with median TVDTs ranging from 2.4 to 7.5 months [10,21,22,23]. A recent meta-analysis, which included 20 studies encompassing 1374 HCCs in 1334 patients, reported a pooled TVDT of 4.6 months (95% CI 3.9 to 5.3) [8]. In a large, multicenter cohort, Rich et al. found that TVDT is significantly associated with several clinical variables, including initial tumor diameter, baseline AFP levels, and the etiology of liver disease [16]. Of their cohort, 25.2% of the patients were classified as having rapid tumor growth, defined as a TVDT of less than 90 days. Similarly, 38.2% of the patients in the current study were classified as having rapid growth, defined as a TVDT of less than or equal to 2.5 months.

The present study found that worse performance status and higher baseline AFP were associated with significantly worse overall survival. Similar results have been previously reported. A multicenter study of HCC patients treated with TACE at Chinese Tertiary hospitals found that worse performance status was an independent predictor of worse OS amongst the whole cohort [24]. A retrospective study of 105 HCC patients treated with TACE at a large North American university-affiliated hospital reported that an ECOG performance score of >0 was an independent prognosticator of worse survival on multivariate analysis [25]. Prior studies have also identified increased AFP as a predictor of worse survival following TACE [26,27], and there may be a genetic correlation with high AFP expression [28].

The presence of more than five hepatic tumors on baseline imaging was found to be a predictor of worse LTPFS. Multinodular HCC is known to be associated with local recurrence after TACE. A study of 287 HCC patients treated with TACE reported that multinodular disease was associated with recurrence even after a complete response on cross-sectional imaging [26]. TACE and TAE may not effectively treat microscopic satellite tumors, leading to recurrence after treatment [21].

Prior studies describing outcomes after TACE are used as a basis for comparison for TAE because experience with TACE is more extensively described in the literature, and both are ischemia-based transarterial therapies with very similar clinical outcomes [8,9]. A prior randomized control trial comparing TACE and TAE found no statistically significant difference in OS, progression-free survival, imaging response, or frequency of adverse events [7]. It is reasonable to assume that the outcomes described in the current study after TAE are likely to be found after TACE.

No single common genetic mutation differentiated between rapidly growing and slowly growing tumors. Based on the TVDT threshold of 2.5 months, nearly 40% of the patients were classified as having rapidly growing tumors, a proportion consistent with prior studies. If a single mutation were associated with rapid tumor growth, it should be present in a sizable proportion of patients. As such, this study only included genetic mutations present in greater than 10% of the patient cohort for subsequent analysis. A prior mouse study found that a TP53 mutation was associated with increased HCC proliferation [29]. However, the current study did not find an association between TP53 and TVDT. The lack of a genetic association suggests that the HCC tumor growth rate may be more nuanced, possibly accounted for by individual genetic signatures, patient-specific epigenetic changes, and environmental factors. The relationship between genetic alterations and a TVDT threshold of 2.5 months was specifically investigated with the goal of identifying a clinically meaningful association. In a larger cohort, the effect of less common mutations and/or combinations of mutations may be assessed for a link with TVDT. Future research may also assess epigenetics, transcriptomics, and other omics data to more comprehensively assess the mechanisms of growth heterogeneity in HCC.

The current study did not find a statistically significant link between the etiology of liver disease and rapid tumor growth. A prior study of 175 East Asian HCC patients demonstrated that the etiology of liver disease affects baseline TVDT, with hepatitis-B-associated HCC exhibiting more rapid tumor growth than hepatitis-C-associated HCC [15]. The current study’s smaller cohort may have been insufficiently powered to detect an association.

A TP53 mutation was a predictor of worse OS and LTPFS on univariate analysis. A TP53 mutation in HCC is linked to a more aggressive disease pattern and more prevalent microvascular invasion, and it may be a prognostic factor, though the extent of existing evidence remains limited [30,31]. The relationship between TP53, overall survival, and local tumor progression after TAE for patients with HCC was investigated in a prior study [15] and is not the focus of the current manuscript.

This study had several limitations. The patient selection criteria for this study led to an inherent selection bias, as only patients with a >30-day interval between initial and follow-up imaging, inclusive of imaging during the TAE procedure, were included. The inclusion of patients with a shorter time to treatment may have led to differences in TVDT and survival outcomes; however, the >30-day interval was deemed necessary to provide sufficient time for measurable tumor growth. Although the results and interpretation are limited by the retrospective nature of the data collection, due to ethical considerations regarding the measurement of untreated tumor growth over time in a human population, this study is difficult to perform prospectively. This study was also limited by its sample size and single-center design, which limits the generalizability of the findings to a broader patient population.

There were multiple limitations involving tumor measurement using imaging. There are inherent differences between diagnostic CT and MRIs compared with intraprocedural cross-sectional imaging, including image quality and phases of acquisition, which may lead to variations in tumor measurement. Cases of poor tumor visualization on intraprocedural imaging were excluded to minimize this difference. For purposes of volume calculations, the tumors were measured with bi-directional trans-axial measurements and assumed to be an oblate spheroid. This simplifying assumption was deemed valid because most of the measured tumors were ellipsoids, and trans-axial measurements of tumors, such as in the RECIST criteria, are an established method of evaluating tumor burden in clinical practice [32]. Methods of estimating TVDT include the Schwartz equation and the specific growth rate [33]. The Schwartz equation was used because it was the best-established and most frequently used method for calculating TVDT in prior studies [10,11,21,22,23]

## 5. Conclusions

In conclusion, patients with HCCs that exhibit rapid growth as per shorter TVDTs may have worse OS following TAE. There does not appear to be a single genetic mutation correlated with rapid tumor growth. The optimal approach for clinical management of patients with rapidly growing HCCs remains unclear and is a topic for future investigation. 

## Figures and Tables

**Figure 1 cancers-17-03346-f001:**
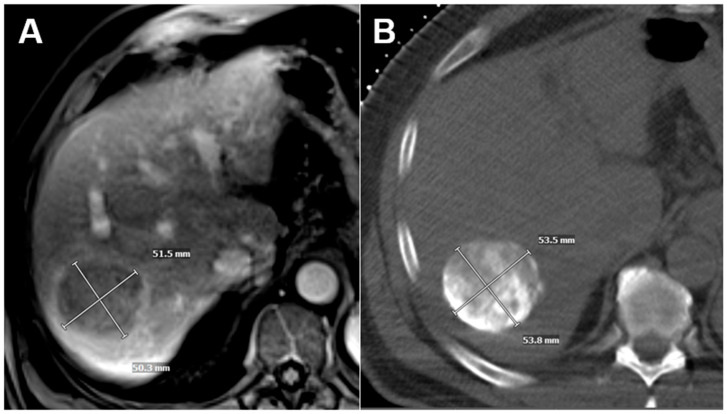
Example of an HCC initially measured on contrast-enhanced MRI (**A**) and then remeasured using tumor staining on intraprocedural non-contrast CT obtained after TAE using a combined angio/CT suite (**B**).

**Figure 2 cancers-17-03346-f002:**
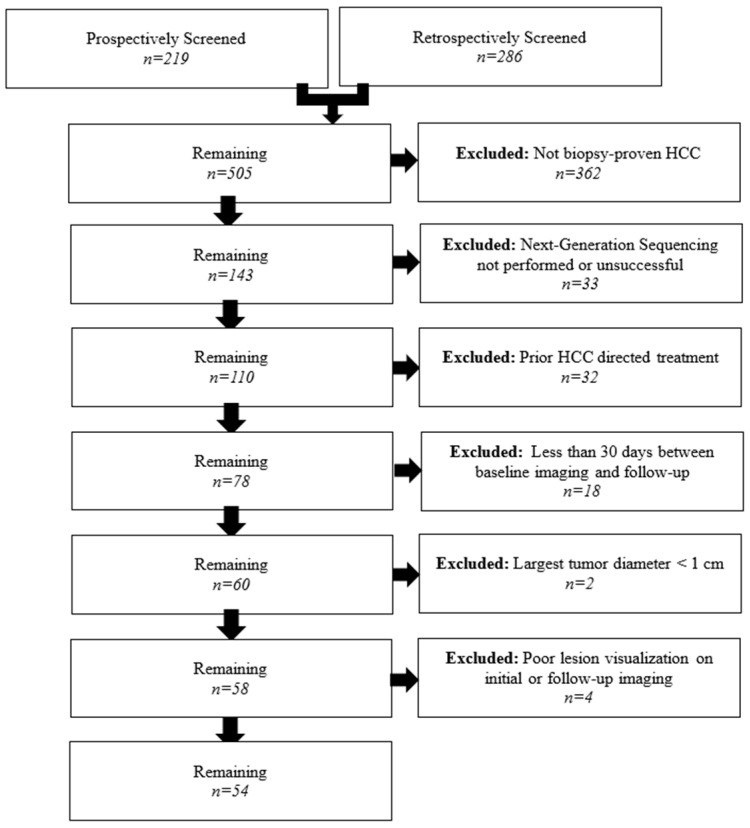
Flowchart of patient inclusion and exclusion for the present study.

**Figure 3 cancers-17-03346-f003:**
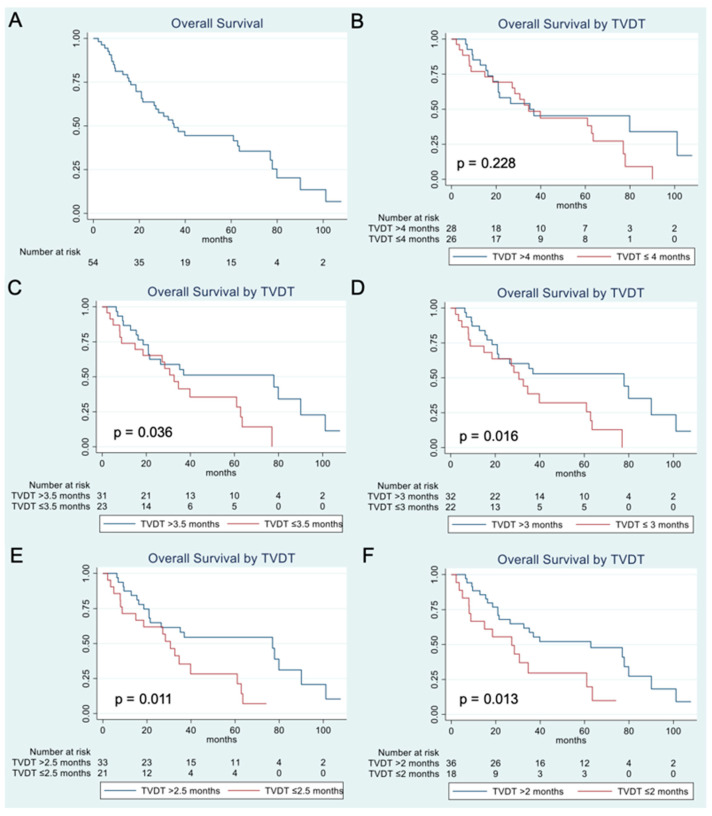
Kaplan–Meier analysis of overall survival for the study population (**A**) and stratified by a tumor volume doubling time threshold of 4 months (**B**), 3.5 months (**C**), 3 months (**D**), 2.5 months (**E**), and 2 months (**F**).

**Figure 4 cancers-17-03346-f004:**
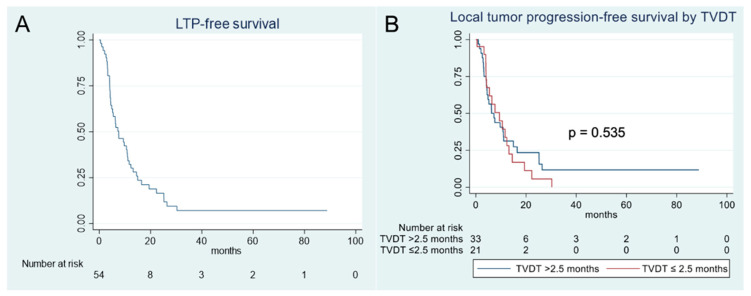
Kaplan–Meier analysis of local tumor progression-free survival for the study population (**A**) and stratified by a tumor volume doubling time threshold of 2.5 months (**B**).

**Figure 5 cancers-17-03346-f005:**
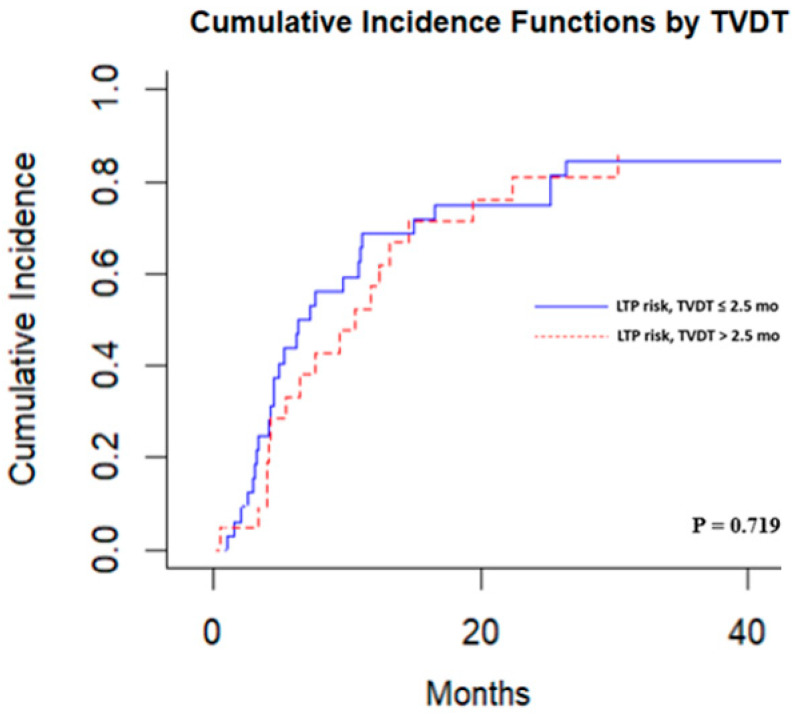
Competing risks regression analysis stratified by a tumor volume doubling time threshold of 2.5 months.

**Figure 6 cancers-17-03346-f006:**
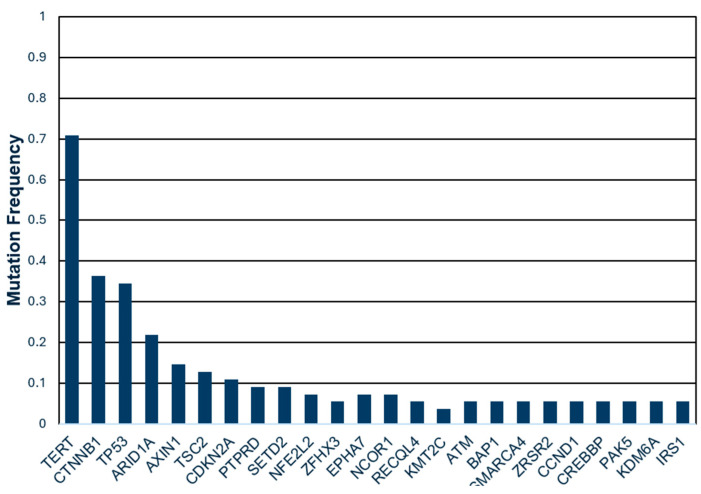
Histogram of genetic mutations present in the study population, as identified with next-generation sequencing.

**Table 1 cancers-17-03346-t001:** Baseline patient demographics and lesion characteristics.

	All Patients n = 54	Rapidly Growing Tumors TVDT ≤ 2.5 Months n = 21	Slowly Growing Tumors (TVDT > 2.5 Months) n = 33	*p*-Value
Age, years (median, IQR)	69 (59–73)	62 (57–71)	70 (63–73)	0.14
Sex				0.29
Male	47 (87%)	17 (81.0%)	30 (90.9%)	
Female	7 (13%)	4 (19%)	3 (9.1%)	
Child Pugh score				0.64
A	50 (92.6%)	19 (90.5%)	31 (93.9%)	
B	4 (7.4%)	2 (9.5%)	2 (6.1%)	
ECOG score				0.52
0	37 (68.5%)	15 (71.4%)	22 (69.7%)	
1 or 2	17 (31.5%)	6 (28.6%)	11 (33.3%)	
Genetic mutations				
TERT	39 (72.2%)	16 (76.2%)	23 (69.7%)	0.60
CTNNB	20 (37%)	7 (33.3%)	13 (39.4%)	0.65
TP53	19 (35.2%)	7 (33.3%)	12 (36.4%)	0.82
ARID1A	12 (22.2%)	4 (19.1%)	8 (24.2%)	0.65
AXIN1	8 (14.8%)	4 (12.1%)	4 (19.1%)	0.49
TSC2	7 (13%)	4 (19.1%)	3 (9.1%)	0.29
CDKN2A	6 (11.1%)	2 (9.5%)	4 (12.2%)	0.77
Liver Disease Etiology ^1^				
NASH	15 (27.8%)	6 (28.6%)	9 (27.3%)	1.0
Alcohol	13 (24.1%)	5 (23.8%)	8 (24.2%)	1.0
HBV	4 (7.4%)	0	4 (12.1%)	0.29
HCV	20 (37%)	10 (47.6%)	10 (30.3%)	0.25
Other	3 (5.6%)	0	3 (9.1%)	0.28
AFP at baseline ^2^				0.69
≤200 ng/mL	43 (79.6%)	16 (76.2%)	27 (81.8%)	
>200 ng/mL	9 (16.7%)	4 (19.0%)	5 (15.2%)	
Extrahepatic disease	6 (11.1%)	2 (9.5%)	4 (12.1%)	0.77
Macrovascular invasion	8 (14.8%)	5 (23.8%)	3 (9.1%)	0.14
Infiltrative pattern	2 (3.7%)	2 (9.5%)	0	0.07
More than five tumors	8 (14.8%)	4 (19.1%)	4 (12.1%)	0.49
Bilobar disease	23 (42.6%)	12 (57.1%)	11 (33.3%)	0.09
Size of index tumor, mm (median, IQR)	52 (33–72)	52 (37–72)	52 (31–71)	0.45

^1^ Etiology of liver disease was unknown for 5 patients, and 5 patients had more than one potential etiology. ^2^ Baseline AFP was not available for 2 patients.

**Table 2 cancers-17-03346-t002:** Overall survival stratified by tumor volume doubling time.

	Overall Survival Stratified by TVDT n = Number of Patients in Each Group, Months (95%CI)		
TVDT Threshold	Rapid Tumor Growth	Slow Tumor Growth	*p*-Value
4 months	n = 26,	n = 28,	0.228
	34.6 months (18.5–63.5)	35.1 months (18.6–101.2)	
3.5 months	n = 23,	n = 31,	0.036
	32.6 months (14.9–61.0)	77.9 months (20.9–90.1)	
3 months	n = 22,	n = 32,	0.016
	30.6 months (8.8–61.0)	77.9 months (21.0–90.1)	
2.5 months	n = 21,	n = 33,	0.011
	30.6 months (8.8–39.8)	77 months (21.0–90.1)	
2 months	n = 18,	n = 36,	0.013
	27.2 months (8.0–61.0)	62.8 months (21.5–80.0)	
1.5 months	n = 12,	n = 42,	0.062
	27.2 months (3.5–61.0)	39.8 months (21.5–80.0)	

**Table 3 cancers-17-03346-t003:** Univariate regression analysis of factors affecting overall survival.

Factor	HR (95% CI)	*p*-Value
Age	1.01 (0.98–1.04)	0.655
Sex		0.180
Male	Reference
Female	1.84 (0.75–4.51)
Child Pugh score		0.054
A	Reference
B	3.33 (0.98–11.26)
ECOG status		0.003

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

**Table 4 cancers-17-03346-t004:** Multivariate regression analysis of factors affecting overall survival.

Factor	HR (95% CI)	*p*-Value
TVDT		0.036
>2.5 months	Reference
≤2.5 months	2.21 (1.05–4.66)
ECOG status		0.006

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

**Table 5 cancers-17-03346-t005:** Univariate regression analysis of factors affecting local tumor progression-free survival.

Factor	HR (95% CI)	*p*-Value
Age	1.02 (0.99–1.05)	0.288
Sex		0.405
Male	Reference
Female	1.59 (0.53–4.72)
Child Pugh score		0.689
A	Reference
B	0.79 (0.24–2.54)
ECOG status		0.036

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

**Table 6 cancers-17-03346-t006:** Multivariate regression analysis of factors affecting local tumor progression-free survival.

Factor	HR (95% CI)	*p*-Value
ECOG score		0.351

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

**Table 7 cancers-17-03346-t007:** Local tumor progression-free survival stratified by tumor volume doubling time.

	Local Progression Free Survival Stratified by TVDT n = Number of Patients in Each Group, Median Months (95%CI)		
TVDT Threshold	Rapid Tumor Growth	Slow Tumor Growth	*p*-Value
4 months	n = 26,	n = 28,	0.922
	9.4 months (4.1–13.2)	6.3 months (4.2–11.0)	
3.5 months	n = 23,	n = 31,	0.239
	7.6 months (4.5–15.0)	7.6 months (4.0–12.4)	
3 months	n = 22,	n = 32,	0.367
	7.6 months (4.0–12.4)	7.2 months (4.4–11.1)	
2.5 months	n = 21,	n = 33,	0.535
	9.4 months (4.1–12.4)	7.2 months (4.2–11.1)	
2 months	n = 18,	n = 36,	0.149
	6.4 months (4.0–12.4)	7.6 months (4.5–11.1)	
1.5 months	n = 12,	n = 42,	
	7.6 months (4.0–13.2)	7.2 months (4.4–11.0)	0.511

## Data Availability

The datasets presented in this article are not readily available because the prospectively collected data are part of an ongoing protocol. Requests for the datasets should be directed to zhaok@mskcc.org.

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
