# Peer review of "Hepatocellular Carcinoma Growth Kinetics and Outcomes After Transarterial Embolization: A Single-Center Analysis"

_cancers, 2025, doi:10.3390/cancers17203346_

Round 1

Reviewer 1 Report

Comments and Suggestions for Authors

The study investigated the relationship between tumor volume doubling time (TVDT) and patient prognosis in hepatocellular carcinoma (HCC) after transarterial embolization (TAE), and analyzed the impact of gene mutations on TVDT. Here are 5 suggestions for this study:
1. Clarify the limitations and consistency of the TVDT calculation method. Tumor measurement is subjective, especially between different imaging modalities (CT, MRI, intraoperative imaging), and the lack of consistency evaluation may affect the accuracy of TVDT.
2. Increase the biological or clinical basis for the "rapid growth" threshold (2.5 months). Explain why 2.5 months was chosen as the threshold for TVDT, and whether it is based on previous literature, statistical distribution (such as median), or clinical consensus.
3. Explore the statistical efficacy of gene mutation analysis. It was explicitly pointed out in the discussion that due to the small sample size (n=54) and low mutation frequency (such as only 6 cases of CDKN2A), the study may lack statistical power to detect the association between mutations and TVDT.
4. Supplement imaging standards for the definition of "local tumor progression", improve the reproducibility and transparency of outcome indicators, and avoid subjective bias.
5. In the conclusion section, it is recommended that future research combine transcriptome, epigenetics, and other omics data to more comprehensively reveal the driving mechanisms of HCC growth heterogeneity.

Author Response

Reviewer 1:

The study investigated the relationship between tumor volume doubling time (TVDT) and patient prognosis in hepatocellular carcinoma (HCC) after transarterial embolization (TAE), and analyzed the impact of gene mutations on TVDT. Here are 5 suggestions for this study:

  1. Clarify the limitations and consistency of the TVDT calculation method. Tumor measurement is subjective, especially between different imaging modalities (CT, MRI, intraoperative imaging), and the lack of consistency evaluation may affect the accuracy of TVDT.

- Thank you. We agree that methods of measuring tumor volume and calculating TVDT are an estimation. Though trans-axial measurements are imperfect, they are an established method of assessing tumor burden in clinical practice, for example the RECIST criteria. We also utilized the Schwartz equation for calculation of TVDT because it is well validated and has been used in several prior studies which assessed TVDT. We have expanded upon the discussion to touch upon these points.

  1. Increase the biological or clinical basis for the "rapid growth" threshold (2.5 months). Explain why 2.5 months was chosen as the threshold for TVDT, and whether it is based on previous literature, statistical distribution (such as median), or clinical consensus.

- Thank you. We expanded upon the methods to clarify that we used a statistical method called the “minimum p-value approach” to determine the optimal threshold for analysis and cited a reference. Within our discussion, we review how the threshold for rapid growth we found was also similar to thresholds established by prior studies.

  1. Explore the statistical efficacy of gene mutation analysis. It was explicitly pointed out in the discussion that due to the small sample size (n=54) and low mutation frequency (such as only 6 cases of CDKN2A), the study may lack statistical power to detect the association between mutations and TVDT.

- Thank you. In this initial exploratory analysis, we only chose to focus on single mutations that were common in our cohort. In a larger cohort, less common mutations, and perhaps combinations of mutations, may be analyzed. The discussion has been expanded to highlight this limitation.

  1. Supplement imaging standards for the definition of "local tumor progression", improve the reproducibility and transparency of outcome indicators, and avoid subjective bias.

- Thank you. We clarified the methods to say that local tumor progression was defined as recurrence or increase in size of enhancing tumor at a site that was previously treated with TAE.

  1. In the conclusion section, it is recommended that future research combine transcriptome, epigenetics, and other omics data to more comprehensively reveal the driving mechanisms of HCC growth heterogeneity.

- Thank you. This is a good point and have included this within the discussion. We feel that while worth mentioning, it is too specific of a point to state in the conclusion.

Reviewer 2 Report

Comments and Suggestions for Authors

This retrospective study explores the growth kinetics of hepatocellular carcinoma (HCC) and clinical outcomes in patients treated with Transarterial Embolization (TAE), based on a single-center cohort. The topic is clinically significant, particularly as embolization remains a key therapy in intermediate-stage HCC and understanding tumor behavior post-TAE is critical for treatment planning and prognosis. The study provides useful insights into tumor growth dynamics, response rates, and time-to-progression post-embolization. However, several areas need clarification or enhancement, particularly in terms of statistical analysis, interpretation of findings, and clinical framing.

Major Comments
1. The manuscript would benefit from a clearer statement of the primary hypothesis or clinical question. Are the authors aiming to validate a predictive model, identify prognostic factors, or describe tumor kinetics?

2. Several key variables need more detailed description:
a. How was tumor growth rate calculated? (e.g., linear, exponential growth model?).
b. What imaging intervals were used to assess growth pre- and post-TAE?
c. Include criteria for patient inclusion/exclusion more explicitly (e.g., ECOG status, comorbidities, prior treatments).

3. The statistical methods section is underdeveloped.
a. Clearly state the tests used for comparing groups (e.g., t-test, Kaplan–Meier, Cox regression).
b. Were multivariate models applied to adjust for confounders (e.g., tumor size, number of lesions, Child-Pugh)?
c. Provide confidence intervals and effect sizes, not just p-values.

4. The Discussion lacks depth in interpreting:
a. The biological relevance of different tumor growth rates.
b. How these findings compare to other embolization cohorts (e.g., TACE or DEB-TACE).
c. Limitations (e.g., retrospective nature, sample size, single-center) should be clearly stated.

5. Suggest discussing how the data could be used to inform clinical decision-making, e.g.:
a. Better patient selection for embolization.
b. Timing of repeat TAE or switch to systemic therapy.

Minor Comments

1. Include growth kinetics graphs (e.g., spider plots or waterfall plots showing tumor size changes over time).
2. Survival plots (e.g., Kaplan–Meier for OS or PFS) are encouraged.
3. Consider summarizing baseline patient characteristics in a dedicated table, with statistical comparison between subgroups.

Author Response

Reviewer 2:

This retrospective study explores the growth kinetics of hepatocellular carcinoma (HCC) and clinical outcomes in patients treated with Transarterial Embolization (TAE), based on a single-center cohort. The topic is clinically significant, particularly as embolization remains a key therapy in intermediate-stage HCC and understanding tumor behavior post-TAE is critical for treatment planning and prognosis. The study provides useful insights into tumor growth dynamics, response rates, and time-to-progression post-embolization. However, several areas need clarification or enhancement, particularly in terms of statistical analysis, interpretation of findings, and clinical framing.

Major Comments
1. The manuscript would benefit from a clearer statement of the primary hypothesis or clinical question. Are the authors aiming to validate a predictive model, identify prognostic factors, or describe tumor kinetics?

- Thank you. We clarified that TVDT may be a prognostic factor for patients with HCC treated with TAE within the introduction and expanded in the conclusion that the optimal approach for management of patients with rapidly growing HCCs is a future direction for research.

2. Several key variables need more detailed description:
a. How was tumor growth rate calculated? (e.g., linear, exponential growth model?).

- Thank you. We describe the calculation of tumor volume doubling time using the Schwartz equation within the methods under subheading “Tumor Volume Doubling Time Calculation”. We have also expanded the discussion regarding calculating of TVDT to highlight that this is a well-established method for calculation of tumor growth rate.

b. What imaging intervals were used to assess growth pre- and post-TAE?

- Thank you. We added within the results that the median time between initial and follow-up imaging was 63.5 (IQR 31-81.75) days.

c. Include criteria for patient inclusion/exclusion more explicitly (e.g., ECOG status, comorbidities, prior treatments).

- Thank you. Agree that inclusion/exclusion criteria can be confusing. We hope that the flowchart of patient inclusion and exclusion for the present study (Figure 2) sufficiently clarifies this.

3. The statistical methods section is underdeveloped.
a. Clearly state the tests used for comparing groups (e.g., t-test, Kaplan–Meier, Cox regression).

- Thank you. We have included a “Statistical Analysis” sub-section within the methods which should clarify. We have also expanded this to more clearly discuss the approach for determining a TVDT threshold.

b. Were multivariate models applied to adjust for confounders (e.g., tumor size, number of lesions, Child-Pugh)?

- Thank you. Yes, we utilized factors that were statistically significant on univariate Cox regression for multivariate Cox regression analysis. This is stated within the “Statistical Analysis” Sub-section of the methods.

c. Provide confidence intervals and effect sizes, not just p-values.

- Thank you. Within our data tables, we include IQR and confidence intervals, when applicable.

4. The Discussion lacks depth in interpreting:
a. The biological relevance of different tumor growth rates.

- Thank you. This study in particular aims to examine associations between TVDT and outcomes after TAE. We have expanded the discussion to mention that future research may also assess epigenetics, transcriptomics, and other omics data to more comprehensively assess the mechanisms of growth heterogeneity in HCC.

b. How these findings compare to other embolization cohorts (e.g., TACE or DEB-TACE).

- Thank you. Within our discussion, many of the prior studies we cite for comparison evaluated HCC patients treated with TACE. We also included a paragraph discussing how TACE and TAE have very similar outcomes after treatment of patient with HCC and hypothesize that the results found in the current study after TAE would likely be found in HCC patients treated with TACE.

c. Limitations (e.g., retrospective nature, sample size, single-center) should be clearly stated.

- Thank you. Within the discussion, we mention the limitations of the study. We have also more clearly stated that it is limited by it’s single-center design and limited sample size.

5. Suggest discussing how the data could be used to inform clinical decision-making, e.g.:
a. Better patient selection for embolization.
b. Timing of repeat TAE or switch to systemic therapy.

- Thank you. This is a great point. We added “The optimal approach for clinical management of patients with rapidly growing HCCs remains unclear and is a topic for future investigation. These patients may benefit from a combination of systemic and liver directed therapies.”

Minor Comments

  1. Include growth kinetics graphs (e.g., spider plots or waterfall plots showing tumor size changes over time).

- Thank you. For this study, we calculated TVDT between two time points, so such a plot of tumor size changes over time cannot be made.

  1. Survival plots (e.g., Kaplan–Meier for OS or PFS) are encouraged.

- Thank you. We have included Figures 3 and 4 which are Kaplan-Meier survival plots

  1. Consider summarizing baseline patient characteristics in a dedicated table, with statistical comparison between subgroups.

- Thank you. Table 1 is a baseline patient demographics and lesion characteristics table with comparison between the subgroups.

Round 2

Reviewer 1 Report

Comments and Suggestions for Authors

The author answered all my questions. Some limitations were also reflected in the discussion. The quality of the revised manuscript has been improved. I am satisfied with the author's work.

Reviewer 2 Report

Comments and Suggestions for Authors

The authors have addressed my comments.